# Bone Marrow Adipocytes—Role in Physiology and Various Nutritional Conditions in Human and Animal Models

**DOI:** 10.3390/nu13051412

**Published:** 2021-04-22

**Authors:** Katarzyna Piotrowska, Maciej Tarnowski

**Affiliations:** Department of Physiology, Pomeranian Medical University in Szczecin, al. Powstancow Wlkp.72, 70-111 Szczecin, Poland; maciejt@pum.edu.pl

**Keywords:** bone marrow adipose tissue, marrow fat, obesity, calorie restriction, dietary interventions, animal models

## Abstract

In recent years, adipose tissue has attracted a lot of attention. It is not only an energy reservoir but also plays important immune, paracrine and endocrine roles. BMAT (bone marrow adipose tissue) is a heterogeneous tissue, found mostly in the medullary canal of the long bones (tibia, femur and humerus), in the vertebrae and iliac crest. Adipogenesis in bone marrow cavities is a consequence of ageing or may accompany pathologies like diabetes mellitus type 1 (T1DM), T2DM, anorexia nervosa, oestrogen and growth hormone deficiencies or impaired haematopoiesis and osteoporosis. This paper focuses on studies concerning BMAT and its physiology in dietary interventions, like obesity in humans and high fat diet in rodent studies; and opposite: *anorexia nervosa* and calorie restriction in animal models.

## 1. Introduction

In recent years, adipose tissue has attracted a lot of attention. It is not only an energy reservoir but also plays important immune, paracrine and endocrine roles [1,2]. Moreover, the rising incidence of worldwide obesity has increased the pace of research. Basically, adipose tissue has been classified into three types: white, brown (WAT and BAT, respectively) and beige adipose tissue. Since the mid-1990s, a third important adipose tissue has received increasing attention—the marrow fat (bone marrow adipose tissue—BMAT). Previously, this fattissue was considered a space filler of the bone marrow (BM) with an unknown origin or function. With further in-depth research, as well as the employment of novel experimental techniques such as lineage tracing, it is now acknowledged that BMAT originates from skeletal lineages, maintaining bone marrow homeostasis and influencing whole-body energy metabolism [3,4]. BMAT is a heterogeneous tissue, found mostly in the medullary canal of the long bones (tibia, femur and humerus), in the vertebrae and iliac crest. Haematopoiesis and osteogenesis are the processes responsible for the formation of bone marrow and bone, respectively. Adipogenesis in bone marrow cavities is a consequence of ageing or may accompany pathologies like diabetes mellitus type 1 (T1DM), T2DM, anorexia nervosa, oestrogen and growth hormone deficiencies or impaired haematopoiesis and osteoporosis.

## 2. Types of Adipose Tissue

Adipose tissue is distributed in distinct depots in the human body. Histologically it is characterised by the presence of adipocytes containing lipid filled vacuoles. Adipose tissue may be further differentiated, mostly based on its origin, function and molecular/biochemical features (reviewed in Li [5]) The subtypes of adipose tissue include WAT, BAT, beige or so-called ‘brite’ (brown/white) adipose tissue and MAT/BMAT.

### 2.1. White Adipose Tissue

Mitochondria-sparse WAT is mostly located in the subcutaneous and visceral depots and functions as lipid storage, which is mobilised and liberates the free fatty acids from triglycerides in situations of energy demand. It constitutes 10–20% of body weight in lean humans [6]. When excess energy exists, lipogenic enzymes are stimulated to synthesise triglycerides for further use [7], while caloric intake restriction triggersthe release of free fatty acids from fat stores via enzymatic lipid hydrolysis into the blood stream [8]. Recent data show important differences between the two WAT depots, as visceral adipocytes are more responsive to lipolytic signals which upregulate the transport of free fatty acids, while subcutaneous adipocytes serve as stable energy reserves [9,10]. In periods of increased calorie intake, WAT expands through adipocyte hypertrophy (cell size increase) and hyperplasia (cell number increase), and terminal peroxisome proliferator-activated receptor delta (PPAR-γ)- driven differentiation of adipocyte progenitors [10]. This expansion is especially significant in obese persons with metabolic syndrome, increased risk of T2DM or cardiometabolic disease [11].

Moreover, it plays an important role as a secretory organ, as it releases cholesterol, steroid hormones and vital molecules involved in immunological (immunomodulation) and endocrine processes, energy balance and regulation of food intake such as interleukin-6(IL-6), adipokines or tumour necrosis factor-α(TNFa). Inflammatory molecule release may be also associated with macrophages infiltrating the tissue and with phagocytosing of dead adipocytes, which produces local inflammation that in future associates with the development of insulin resistance, particularly in obesity [12,13].

### 2.2. Brown Adipose Tissue

Brown adipocytes originate from myogenic lineage (dermomyotomes expressing MYF-5) [14]. BAT plays an important role in the control of body temperature. The main function of BAT is transfer energy from food into heat and the activity of this tissue is under the control of norepinephrine released from sympathetic nerves [15]. In foetuses and newborns, BAT, by intensive metabolism of fatty acids and expression of mitochondrial uncoupling protein 1 (UCP1), is responsible for heat production (thermogenesis). As the body develops, the presence of this tissue is limited and in adults it is barely detectable [16]. Some amount of cold-activated mitochondria-enriched BAT is present as discrete tissue deposits above the clavicle and in the subscapular region of the back [16,17,18]. Along with its role in adaptive thermogenesis, BAT seems to have an important role in regulating insulin resistance, protecting against obesity and diabetes [5,9,19,20]. BAT is inversely correlated with the body mass index [18].

Beige adipocytes (the‘third’ fat tissue) are an intermediate type of the above-mentioned adipocyte cell populations. They are mostly found in WAT depots and responds to stimuli like cold, the sympathetic nervous system (via catecholamines and β-adrenergic signalling), thyroid hormones or exercise. Beige adipocytes express UCP-1 protein at a lower level than brown adipocytes but are more sensitive to stimulation (reviewed in Martinez-Fernandez [20] and Chechi [21]).

The main subject of this review is, however, BMAT.

### 2.3. Bone Marrow Adipose Tissue

Presently, with the help of the recent advent of non-invasive methods to measure and analyse the marrow adipose tissue, the role of BMAT is receiving greater appreciation as it is a very important element of the marrow microenvironment, making up ~50–70% of the marrow volume. This adipose tissue is also known as marrow adipose tissue (MAT) or yellow adipose tissue. It is an adipose depot with unique features distinguishing it from the better characterised extramedullary sites. It accounts for approximately 5–10% of the total fat mass in healthy, lean, adult humans (reviewed in Fazeli [22]). Application of MRI-based techniques demonstratesthat the amount of total BMAT in a skeleton of average size ranges from 0.5 to 3 kg [23].

Bone marrow adipocytes originate in the BM from mesenchymal stem cells (MSC). The key transcriptional factors, PPARγ and c/EBPα, control BM adipogenesis [9]. BM MSCs have the potential to differentiate towards osteoblasts, adipocytes or chondrocyte lineages [24,25,26]. PPARγ insufficiency in BM progenitor cells leads to osteoblastogenesis, which is considered a competitive process for adipogenesis [27]. The two processes are strictly and opposingly regulated. Factors that promote osteogenesis like mechanical forces, growth hormone or insulin-like growth factor 1 (IGF-1) limit adipogenesis and vice versa, pro-adipogenic stimuli like oxidative stress, immobilisation, elevated glucocorticoid levels limit osteoblastogenesis [4,25,28]. Thus, the BM MSCs shift towards either lineage results from a complex interplay of systemic and local mediators.

For many years, this tissue was considered as only a filler of trabecular bone cavities (ribs, sternum, vertebrae) and the medullary canal of long bones (tibia, femur, humerus) involved in the transition of red (haematopoietic) bone marrow to yellow (non-haematopoietic) bone marrow. It gradually accumulates in areas of trabecular bone of the femur, tibia and vertebrae and fills the entire marrow cavity by the third decade of human life [22,28], with males demonstrating greater amounts of BMAT compared to females [29]. In the red marrow, where haematopoiesis and bone remodelling are active, adipocytes are less frequent and account for up to 45% of the marrow, while in the yellow marrow, where haematopoiesis is almost absent, adipocytes are densely packed and fill up to 90% of the marrow compartment [5]. It has also been shown that enlargement of this compartment accompanies metabolic diseases, states of increased bone fragility and obesity [30].

Histologically, BMAT resembles white adipocytes, however it is a heterogeneous population of cells with distinct metabolisms and lipid compositions that, with ageing, gradually replaces the nucleus with lipid droplets [31]. The lipid content of BMAT, which is composed of saturated, monounsaturated and polyunsaturated fat, is used mostly as an energy source for populations of osteoblasts, osteoclasts and haematopoietic cells [32].

What is interesting is that BMAT expresses brown adipocyte gene markers (*Prdm16*, *Dio2* and *PGC1a*), which decrease with ageing and diabetes [33]. Further studies have evidenced that BMAT has a mixed BAT/WAT phenotype. When bone marrow adipocytes were treated with triiodothyronine or a thyroid hormone receptor beta-specific agonist (GC-1) or rosiglitazone, a synthetic agonist for adipocyte-specific PPARγ and a potent insulin sensitiser significantly increased both BAT (*Ucp1*, *Pgc1α*, *Dio2*, *β3AR*, *Prdm16* and *FoxC2*) and WAT (*Adipoq* and *Lep*) markers [33,34,35].

In both rodents and humans, two distinct subtypes of BM adipocytes have been described: constitutive BMAT (cBMAT/cMAT) and regulated BMAT (rBMAT/rMAT) [35]. rBMAT/rMAT is present within active haematopoietic sites such as the mid- to proximal tibia, femur and lumbar vertebrae and develops throughout life. cBMAT/cMAT is located in the distal tibia and caudal vertebrae and develops rapidly after birth [35]. There are some important differences between these two populations: compared with cBMAT/cMAT adipocytes, rBMAT/rMAT adipocytes contain more saturated fatty acids and express lower levels of the adipogenic transcription factors Cebpa and Cebpb, similar to WAT [35].

Furthermore, their response to physiological stress differs: rBMAT/rMAT adipocytes reduce in size and number after cold exposition [35], fasting [36] or prolonged exercise [37]. Additionally, the size of rBMAT/rMAT cells increases with ageing, a high fat diet, caloric restriction and anorexia, irradiation, or treatments with hypoglycaemic thiazolidinediones (insulin-mimetic drugs used for type 2 diabetes treatment) and hyperglycaemia-causing glucocorticoids [38,39]. Conversely, cBMAT/cMAT changes in size in response to external stimuli or pathophysiological changes are less evident [26,35].

## 3. BMAT—Function

The exact function of BMAT remains unclear. When WAT capacity is exhausted, ectopic fat accumulation in other tissue as liver is found, and is regulated by insulin and responds to the energy balance in the body. BMAT is enclosed in cavities of the skeleton and also responds to insulin signalling-triggering molecules (i.e., rosiglitazone, a thiazolidinedione) [40]. As BMAT adipocytes are situated in a unique microenvironment, surrounded by haematopoietic and skeletal lineage cells, this likely contributes to its differential regulation and points to the local microenvironment, in addition to endocrine mediators, as major regulators of BMAT function.

As previously mentioned, both calorie restriction (CR) [2,41] and high fat diets (HFD) anorexia nervosa [38] can increase skeletal BMAT in mice and humans. Research suggests that increased BMAT during calorie restriction is a result of adipogenesis and an increase of adipocyte number rather than an increase in adipocyte size; this may cause uncoupling between BMAT and energy metabolism [2]. Experiments on a mouse model of T1DM indicate that hypoinsulinaemia is insufficient to stop BMAT accumulation [42]. This may indicate different functions of adipocytes in BMAT: while some cells are connected with systemic energy demands, other may be responsible for supporting neighbouring cells.

### 3.1. BMAT and Haematopoiesis

The supporting function, both mechanical and biochemical, of BMAT adipocytes may be envisioned by certain examples. There are data showing BMAT’s contribution to the mechanical properties of the skeleton or its association with haematopoiesis [43,44,45]. In vitro experiments showed increased differentiation of CD34+ cells in the presence of human BMAT and in the same context, in vivo results showed inhibitory characteristics of BMAT, induction of quiescence and loss of haematopoietic progenitor cells [43,46]. Inhibition of BMAT expansion improved haematopoietic engraftment and recovery in an experiment with marrow transplant after bone marrow irradiation in mice [46]. On the other hand, thiazolidinedione (TZD)-induced BMAT overgrowth did not cause a change in progenitor cell number in bone marrow in mice [47]. The latter may be explained by the existence of more complex regulatory loops when PPARs are activated. Furthermore, BMAT may serve as a certain source of cytokines, as discussed later and thus, influenceshaematopoiesis [48]. BMAT may protect osteoblasts from lipotoxicity, by creating a storage space for ectopic lipids [49]. During triglyceride excess, liver and skeletal muscle also store high amounts of lipids [50]. Cells of BMAT harvested from healthy animals show expression of transcription factors connected to adipogenesis: Pparg, Cebpa, Cebpbin [43,51].

### 3.2. Cytokine Production

Research on human adipocytes derived from sternal bone marrow stem cells (BMSCs) show secretion of cytokines IL-6, MIP-1α, G-CSF, and GM-CSF [52]. Adipocytes from the human iliac crest also secreted IL-6 and G-CSF and additionally, IL-8 after seven-days ceiling culture [43].

In a study by Laharrague, it was noted that primary culture of medullary adipocytes, secrete low amounts of IL-1 beta and TNF-alpha, but they secrete significant and regulated levels of IL-6 [53]. Murine BMSC-derived adipocytes produceCXCL1 and CXCL2 [54]. Moreover, genes encoding for IL-6, IL-1β and TNFa were found to be highly expressed in mouse BM adipocytes when compared to epididymal adipocytes, and what is interesting is that the expression of some of these factors was reduced with age [55]. A recent study showed that inflammatory cytokine levels were increased in visceral adipocytes, in high fat-induced obese mice, but BMAT did not exhibit a pro-inflammatory phenotype [56]. Cytokines produced in vitro and ex vivo are shown in Figure 1.

Osteoclast differentiation and activation are stimulated by cytokine secretion and it may seem that the cytokines released from BMAT may be greatly involved in the process [57,58]. Thus, BMAT may havean important role in bone remodelling by contributing to bone loss in osteoporosis and bone destruction during skeletal metastasis [57,58,59]. Adipocytes derived from BMAT also secrete adiponectin (explants of BMAT) and leptin (in vitroisolated cells) [43,53]. Adiponectin promotes insulin sensitivity, fat oxidation and anti-inflammatory action. The main source of adiponectin in normal conditions is WAT. However, during CR, expanding BMAT becomes an additional source of increased adiponectin level [60].

### 3.3. BMAT and the Endocrine System

BMAT quantity changes in metabolic diseases like diabetes, ovarian dysfunctions, obesity and anorexia nervosa (Table 1). BMAT formation is influenced by endocrine factors (growth hormone, oestrogens, glucocorticoids) and is prone to endocrine modification. Depending on the metabolic context, the BMAT compartment may be either increased or decreased.Pituitary-derived growth hormone reduces BMAT formation [61]. The experimental removal of rats’ pituitary glands caused a massive increase of BMAT that was reversed by the administration of somatotropin, but parathormone (PTH), 17β-oestradiol or IGF-1 were not effective [61]. A decreased oestrogen level found in postmenopausal women or in experimental ovariectomy in animal models, FGF-21 and increased level of glucocorticoids also contribute to BMAT expansion [62,63,64,65]. BMAT accumulation may be the result of glucocorticoid increases during CR and short-term transdermal oestrogen administration diminishes vertebral BMAT in premenopausalanorexia nervosa (AN) patients [66]. For example, PTH administration in CR mice induces adipocyte number or adipocyte size reduction in BMAT depending on the experimental protocol [67] and erythropoietin (EPO) administration to female mice reduces BMAT with no effect on total body fat mass [68]. Leptin is noteworthy, and hypoleptinaemia is proposed to promote BMAT accumulation. A decrease in circulating leptin by pituitary removal or WAT loss in CR caused increased BMAT formation [61,69]. On the contrary, intracerebroventricular (ICV) leptin administration resulted in a BMAT decrease in rodents [70,71,72]. The effective loss of BMAT with exogenous leptin administration requires usage of supra-physiological dosages and it is still not estimated if a physiological amount of leptin causes BMAT changes. In conditions of a decreased oestrogen level, BMAT accumulation proceeded in spite of an increased leptin level, which shows that higher leptin alone is not sufficient to regulate BMAT [73,74]. In rodent CR studies, rabbits exhibited hypoleptinaemia without BMAT accumulation, while in female mice, CR caused BMAT formation but without signs of hypoleptinaemia [69]. This may lead to the conclusion that hypoleptinaemia is not necessary nor sufficient for BMAT accumulation. Peripheral administration of exogenous leptin causes a drop in BMAT differentiation [75]. Leptin, probably, causes sympathetic nervous stimulation (SNS). Sympathetic stimulation leads to an energy deficit by nutrient reservoir depletion [76]. The SNS may further trigger a release of norepinephrine and induce, through β-adrenergic receptors, lipolysis in BMAT. Some data indicate that apoptosis mayalso lead to BMAT loss as a result of leptin action [71,77].

### 3.4. BMAT in Metabolic Disorders

Despite its importance, the role of BMAT formation and differentiation in metabolic disorders remainsrelatively unidentified.The increase of BMAT in metabolic disease studies is higher in rodent models than in human studies. In mice with T1DM, the increase of BMAT is not equal in the skeleton;an expansion of BMAT is observed in bones of limbs, but not in the axial skeleton [42,78]. However, in humans with T1DM, changes in BMAT were not detected [78] (Table 1).

A similar observation comes from experiments on mice.In a model of T2DM (ob/ob) and mice fed a high fat diet (HFD), there was a significant increase in BMAT in both [77]. In humans with T2DM, only a minimal or no change in BMAT has been noted [79,80]. In obesity experiments, fed mice exhibited a higher level of BMAT expansion than obese humans [38,81,82]. These differences may be a result of a generally higher BMAT level in humans than in mice, relative to the body mass and also longevity of fat tissue [57,83,84,85].

BMAT expansion may rely on the presence of local and systemic factors regulating BMAT biogenesis and role. Clinical evidence indicates a potential role of circulating triglycerides in BMAT formation [78,86]. In this light, treatment with statins (decrease serum lipids levels) may potentially ‘trim’ BMAT. Additionally, other endocrine factors may also have a role in BMAT expansion.

## 4. BMAT in Dietary Regimes

Changes in quantity and quality of BMAT are observed in different nutritional states. Malnutrition or dietary restrictions and also HFD influence BMAT in a species-, strain- and gender-dependent manner.

### 4.1. BMAT in Obesity and High Fat Diets

In experiments with a HFD, changes in BMAT were also dependent on a type of dietary source of fat used in the study (Table 2) [38,56,87,88,89,90]. In experiments with lard as the main calorie source, BMAT was increased, even if in some studies bone mass remained unchanged [56,89]. An increase of BMAT was observed in male and female C57BL/6 mice, but in females, HFD effects were dependent on the amount of food consumed; female C57BL/6 mice overeating during experiments: in FVB/NJ females which did not overeat the fat mass and BMAT were unchanged [88]. Difference in feeding behaviour are due to genetic background of these strains [88]. A strain-dependent effect of HFD was also noted in experiments with fish oil and saffloweroil on C57BL/6 and C3H/HeJ mice strains [89]. In C3H/HeJ mice, BMAT and osteoclast number in the spine were increased in fish oil fed animals [89]. In studies with fish oil and conjugated linoleic acid, a decrease in total fat content and BMAT amount was observed. Fish oil itself or vegetable oil (corn oil) did not cause a similar effect [90].

The amount of BMAT was also studied in obesity and diabetes in humans. In prepubertal girls, BMAT increases with age, and is correlated with body mass and body fat [91]. In adults, increased body weight is not always associated with increased BMAT in obese and diabetic patients [92]. A positive correlation is observed between BMAT and HbA1c, but not with HOMA-IR (marker of insulin resistance), insulin level, Pre adipocyte factor 1 (Pref-1) or leptin [92,93,94]. On the other hand, weight loss in clinically obese patients decreases the BMAT amount [95]. After gastric bypass, all obese patients lost weight and BMAT. Interestingly, patients with a lower content of BMAT in preoperative examination, lost less BMAT in comparison to patients with a higher BMAT, who lost more BMAT in the post-operative period. Females lost more than males, probably due to endogenous oestradiol levels [95]. However, the BMAT reduction after bariatric procedures depends on the type of procedure used [96]. In a comparison of Roux-en-Y gastric bypass (RYGB) and sleeve gastrectomy (SG) patients, greater loss of weight was observed in RYGB patients than SG. BMAT increased in SG patients in comparison to RYGB patients, probably due to the smaller weight loss in SG [96].

### 4.2. Dietary Restrictions and BMAT

Dietary restrictions are considered beneficial for health and longevity [97]. The impact of caloric restriction has been studied in many animal models, also in the context of bone health.

In human studies with dietary restrictions in the context of BMAT, work has focused mainly on anorexia nervosa (AN) patients, as bone quality is severely lowered and BMAT is increased and may comprise about 31% of total body adipose tissue [2,98]. The differences observed by various groups are mainly due to age and disease severity. Red to yellow marrow transition is related to the degree of nutritional deprivation [99]. In adolescent girls (mild to moderate AN), BMAT shows a strong association with age: in younger girls (<16 years of age) authors observed a positive correlation between BMAT and body mass index (BMI) and bone mineral density (BMD). In older girls (>17 years of age), BMAT was inversely correlated with BMI and BMD [99]. In another study, the authors showed increased BMAT and lowered BMD in bones of the spine and hips as weight-bearing bones, and also in the whole body in adolescent patients. BMAT showed a negative association with bone strength [100]. In older AN patients (>25years of age), BMAT was also increased and BMD was lowered. It was correlated with a decreased leptin level, increased adiponectin level, and interestingly, inversely correlated with HOMA-IR factor and Pref-1 [22,101]. In this age group of AN patients, weight gain was associated with an increase in BMAT in the femur with a % change in BMAT positively associated with a % change in leptin level and positively associated with a % change in subcutaneous adipose tissue [101]. In severe cases of AN, the gelatinous transformation of bone marrow was noted as a sign of starvation [102]. Bone marrow biopsies of some AN patients revealed atrophy of adiposecells and loss of haematopoietic cells, which may be related to fat cell depletion due to catabolism in the tissue [103]. This symptom, however, is rare—0.2–4.8% of all AN patients and is usually present in young males [103]. After weight gain, the restoration of haematopoietic cells and BMAT cells was observed [103].

In animal models of caloric restriction (CR), the effect of food deprivation on BMAT is species-, strain- and gender-specific (Table 3).

In rabbits, a decreased adipocyte size was observed during extensive CR in growing animals, in mature rabbits during 30% CR, BMAT remained unchanged [69]. In the same study, moderate (30%) CR in mice caused an increase of BMAT deposits in the proximal tibia in males but in females the body fat was unchanged, while rBMAT increased. The authors concluded that glucocorticoids were responsible for BMAT expansion, as these hormones were increased in mice and unchanged in rabbits [69]. In growing mice (C57BL/6), 30% of CR caused growth retardation, decreased bone quality (more pronounced in limb bones than in vertebrae), increased bone resorption and dramatically increased BMAT (after 9 weeks of treatment +794%) [41]. In short experiment in C57BL/6 and apo-/- (adiponectin knockout) mice CR negatively affected the microstructure of long bones and vertebrae of C57BL/6 mice but not bones of apo-/- mice [104]. Adiponectin levels in C57BL/6 mice was increased in all measured adipose tissue regions, suggesting that adiponectin is responsible for bone loss during CR [104]. In our experiment with long-term every-other-day feeding (effects similar to about 40% restriction), we found increased BMAT but only in female C57BL/6 mice [105]. In this experiment, bone mineral content measured directly in long bone ashes was unchanged [105]. Short (48h) fasting in rats caused a decrease of BMAT in the tibia but not in the vertebrae of males and females [36]. In long experiment with 40% CR in mature rats increase of BMAT in limb bones was observed [107]. In old rats on CR, serum leptin was on a similar level to young (2-month-old) rats and significantly lower than age matched ad libitum(AL) fed animals [107]. Decreased calorie intake effects on bone and BMAT may be moderated by other interventions, i.e., physical exercise or osteoanabolic agent administration. In an experiment with voluntary running female mice, trabecular bone parameters were decreased in CR running animals, but BMAT in long bones in CR runners was lower than in sedentary CR mice [106]. These authors concluded that lipid storage in BMAT may be a result of different expression of fatty acid (FA) uptake during CR and exercise, as CD36 (marker of FA uptake) was increased in sedentary CR mice in comparison to CR runners and AL groups of animals [106]. In a study with CR and PTH administration in female mice, the effects of hormone’s action was dependent on time, when PTH treatment was introduced to the study protocol. Intervention with PTH when CR was started resulted in the upregulation of osteogenic genes’ expression, but not adipogenesis-related genes—resulting in decreased number of adipocytes in BMAT. Injections with PTH, started after CR was introduced to animals, did not prevent adipogenesis, but caused a reduction of adipocyte size due to lipolysis [67].

## 5. Conclusions

In conclusion, the proper amount of BMAT is required for bone health and an increase in marrow adiposity negatively influences bone quality. Additionally, there is a link between total body fat (TBF) and BMAT. The existing data show a U-shaped association between TBF and BMAT. During conditions of high and low TBF, BMAT is elevated, and normalises with the normalisation of TBF [95]. Studies in AN patients indicate that elevated levels of preadipocyte factor-1 (Pref-1), cortisol, early B-cell factor-1 (Ebf-1) may be determinants of BMAT development [22]. However further research is needed to establish correlation between these factors and BMAT. 

## Figures and Tables

**Figure 1 nutrients-13-01412-f001:**
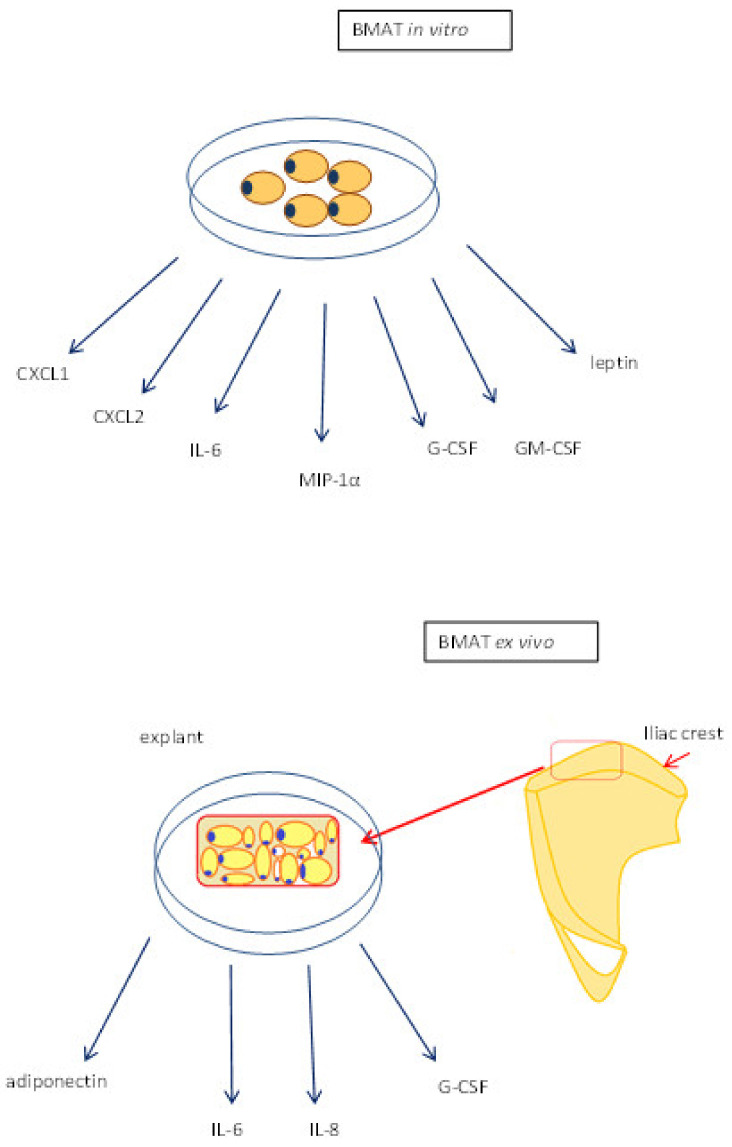
BMAT secretion of cytokines in different experimental protocols (extracted cells or whole tissue explants—without separation of single adipocytes). CXCL1—chemokine (C-X-C motif) ligand 1; CXCL2—chemokine (C-X-C motif) ligand 2; G-CSF—granulocyte colony-stimulating factor; GM-CSF—granulocyte-macrophage colony-stimulating factor; IL-6- interleukin 6; IL-8—interleukin 8; MIP-1α—macrophage inflammatory protein alpha.

**Table 1 nutrients-13-01412-t001:** BMAT in various conditions [66,67,68,78].

Increased BMAT	Decreased BMAT
Ageing	Weight loss (gastric bypass)
Diet regimes (HFD, CR)	Oestrogen administration
Glucocorticoids	PTH administration (and Scl-Ab) (mice rats)
T1DM (mice), T2DM (mice)	EPO administration (mice)
Decreased oestrogen (ovariectomy—mice and rats)	Vitamin D3 administration
Anorexia nervosa	GH, IGF-1
obesity	

HFD—high fat diet; CR—caloric restriction; Scl-Ab—sclerotin-neutralising antibodies; EPO—erythropoietin.

**Table 2 nutrients-13-01412-t002:** Differences in BMAT due to different types of fat in the diet and strain of animals used in the study.

Species/ Strain	Age/Gender	Study Length	Type of Fat	Effects in Bones	Reference
Mice C57BL/6J	12 months females	6 months	linoleic acid + fish oil	↑BMD, ↓BMAT, ↓total fat mass	[90]
			corn oil	↑BMAT	
Mice C57BL/6J	6 weeks males	12 weeks	62% lard	↓cortical bone cross-section area, ↑BMAT, ↓BMD	[87]
Mice C57BL/6J C3H-6T	10 weeks females	sacrificed in 12th month of age	22% fish oil	↑BMAT in spine, ↓BMD more pronounced in 6T strain than in B6,	[89]
			22% safflower oil	prevented weight gain and bone loss in spine	
Mice C57BL/6J	3 weeks males	12 weeks	60% lard	↑BMAT volume, ↑total body weight, ≈ BMD	[38]
	males and females	short term 2 weeks	58% lard	↑BMAT, ↑body weight	
Mice C57BL/6J	3 weeks females	3,8,17 weeks	39% lard+ 6% soybean oil	↑BMAT, ↑body weight, ↓ trabecular bone architecture, ↓femoral cortical bone acquisition	[88]
Mice FVB				≈ BMAT, body weight,	
Mice C57Bl/6J	8 weeks males	12 and 20 weeks	6% fat from lard	↑BMAT volume, ↓BMD, ↓ amount of stem cells in bone marrow	[56]

BMD—bone mineral density.

**Table 3 nutrients-13-01412-t003:** CR in animal models.

Species/ Strain	Age/ Gender	Study Length	Type of Restriction	Effects in Bones	Reference
Mice C57BL/6J Apo-/-	11 weeks males	4–12 weeks	30% CR	↓bone quality, ↓BM, ↑adiponectin in BMAT No influence of CR in apo-/-	[104]
Mice C57BL/6J	3 weeks males	3–9 weeks	30% CR	↓trabecular volume, leptin,↓ osteoblast number,↑trabecular separation, ↑ bone resorption, ↑BMAT	[41]
Mice C57BL/6J	9 weeks males and females	6 weeks	30% CR	Males: ↓ BW,↓leptin,↑rBMAT (tibia), ↑adiponectin, ↑Glucocorticoids Females: ≈TBF, ≈leptin, ↑rBMAT (tibia),	[69]
Rabbits	6 weeks males	7 weeks	30% CR	≈adiponectin,↓BW,↓leptin, ↓bone quality, ≈BMAT	
Mice C57BL/6J	4 weeks Males and females	9 months	every-other day feeding	↑BMAT in females, ≈mineral content in long bones	[105]
Mice C57BL/6J	11 weeks females	6 weeks + voluntary running	30% CR	↓bone quality, ↑BMAT (femurs), ↑CD36	[106]
Sprague- Downey rats	8 months of age	12 months	40% CR	↓ BW, ↓bone quality, ↓leptin, ↑rBMAT(tibia),	[107]
Sprague- Downey rats	Males and females	48h	fasting	↓body mass, ↓BMAT in tibia	[36]

TBF—total body fat; BW—body weight.

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
