# Peer review of "Bone Marrow Adipocytes—Role in Physiology and Various Nutritional Conditions in Human and Animal Models"

_nutrients, 2021, doi:10.3390/nu13051412_

Round 1

Reviewer 1 Report

This Review is focused on characterization of bone marrow adipocytes and their role in physiology and various nutritional conditions in human and animal models. At the beginning of manuscript authors describe different types of adipocytes including bone marrow adipose tissue (BMAT). The next part of manuscript is focused on BMAT function, its connection to metabolic disorders and on dietary effect on BMAT.

Specific comments:

  • In introduction you describe two types of adipose tissue, BAT and WAT. I recommend adding brite/beige adipocytes as the third type of adipocytes because of the origin of breite/beide cells is still in discussion.
  • Abbreviations BMAT and MAT mean the same type adipose tissue. Please use in text only one type of abbreviation
  • Line 49 - I recommend to specify the statement: “subcutaneous and visceral depots and functions as lipid storage, which is mobilized specifically in the fasting state”. Lipid storage serves as energy storage for energy demanding state such as fasting, cold exposure etc. After activation of lipolysis, fatty acids and glycerol are released and used as a fuel for other tissues.
  • Line 50 – 10% WAT of body weight in lean humans. I recommend correct the value 10% - 20% according published data “Normal values for fat mass are 9–18% in males and 14–28% in females” (Hausman, Obesity Reviews, 2001)
  • Line 59 - I recommend to add “obese” persons
  • Line 70 – please specify statement: “Brown adipocytes ………… and become active upon cold exposure.” BAT plays an important role in the control of body temperature. The main function of BAT is transfer energy from food into heat and the activity of this tissue is under the control of norepinephrine released from sympathetic nerves (B, Cannon, 2004).
  • Please specify statement: line 145 “WAT can increase seemingly without 145 limit ….” - when WAT capacity is exhausted, ectopic fat accumulation in other tissue as liver is found.
  • Please specify figure description
  • In Table 1 T2DM increase BMAT but in text you mentioned: “ In humans with T2DM, only a minimal or no change in MAT has been noted”. I recommend add only in mice and also add in the Table1 that obesity increase BMAT.
  • Line 285 – please specify statement:” In experiments with a HFD, changes in BMAT were also dependent on a type of fat used in the study” Do you mean fatty acid composition or composition of the diet?
  • Line 290 – please specify statement: “Female C57BL/6 mice overeating during experiments: in FVB/NJ females which not overeat and the fat mass and BMAT were unchanged”. Did they check pair-fed group? What about difference in genetic background and obesity sensitivity in these mice?
  • Is it known or an explanation why two opposite energy state such as obesity and caloric restriction (anorexia nervosa) cause increase BMAT?

Other comments:

  • description of abbreviation BMAT is missing in abstract
  • Keywords: bone marrow adipose tissue is the same as abbreviation BMAT, please choose only one possibility
  • Please check the spaces between individual words and between sentences.
  • Please check all abbreviations and its description in the text. It is necessary to use for the first time description with abbreviation before using  only  abbreviation
  • Please do not use abbreviation in the title 2.3. but use established abbreviation BMAT in line 85
  • Please correct the spelling in line 155, 322, 341, 337, 340
  • Please add the description: Line 240 - Epo and line 371 - AL
  • Correct, please description of mice C57Bl/6 on page 9.
  • Please check and improve design of References

Reviewer 2 Report

This is an informative and comprehensive review of bone marrow adipocytes from Piotrowska and Tarnowski.  Comments and suggestions are minor as below.

  1. Line 91: There is yellow and red MAT. Yellow adipose tissue is not a common name.
  2. Line 97: The role of PPARg in BM adipogenesis has been well established but less is known about C/EBPa. Please add the reference.
  3. Line 111: The red marrow and yellow marrow likely refer to rMAT and cMAT. Plase make the distinction here to avoid confusion further.
  4. Line 130: cMAT and rMAT are more commonly used.
  5. Line 137-141: this paragraph is confusing. Please modify.
  6. Line 142: cMAT does change in response to external stimuli its just less responsive (ex. Scheller et al Nat Comm 2005).
  7. Line 166-168: Please add reference.
  8. Line 235: A decreased oestrogen level, FGF-21 and glucocorticoids also contribute to MAT expansion. This sentence is unclear.
  9. Line 252-259: The order of the sentences is not fluent, please consider rewording.
